# Stereodefined polymetalloid alkenes synthesis via stereoselective boron-masking of polyborylated alkenes

Nadim Eghbarieh[1], Nicole Hanania[1] & Ahmad Masarwa [1] ✉

Polyborylated-alkenes are valuable polymetalloid reagents in modern organic synthesis, providing access to a wide array of transformations, including the construction of multiple C−C and C−heteroatom bonds. However, because they contain similar boryl groups, many times their transformation faces the main challenge in controlling the chemo-, regio- and stereoselectivity. One way to overcome these limitations is by installing different boron groups that can provide an opportunity to tune their reactivity toward better chemo-, regio- and stereoselectivity. Yet, the preparation of polyborylated-alkenes containing different boryl groups has been rare. Herein we report concise, highly site-selective, and stereoselective boron-masking strategies of polyborylated alkenes. This is achieved by designed stereoselective trifluorination and MIDA-ation reactions of readily available starting polyborylated alkenes. Additionally, the trifluoroborylated-alkenes undergo a stereospecific interconversion to Bdan-alkenes. These transition-metal free reactions provide a general and efficient method for the conversion of polyborylated alkenes to access 1,1-di-, 1,2-di-, 1,1,2-tris-(borylated) alkenes containing $BF_3M$, Bdan, and BMIDA, a family of compounds that currently lack efficient synthetic access. Moreover, tetraborylethene undergoes the metal-free MIDA-ation reaction to provide the mono BMIDA tetraboryl alkene selectively. The mixed polyborylalkenes are then demonstrated to be useful in selective C−C and C−heteroatom bond-forming reactions. Given its simplicity and versatility, these stereoselective boron-masking approaches hold great promise for organoboron synthesis and will result in more transformations.

Organoboron compounds are members of the organometallic family[1,2]. For more than a half-century, organoboron species have been proven to be among the most indispensable diverse classes of reagents in organic synthesis, providing access to valuable chemicals[3–6]. These include their involvement as key intermediates for the synthesis of bioactive materials, natural products, agrochemicals, pharmaceuticals, polymers, and many other important materials[5,7–15]. Their popularity stems from the diverse C−B bond

reactivity profile, which allows the formation of new C−C and C−heteroatom bonds; their non-toxic nature; and their excellent functional-group tolerance characteristics[2,16–24].

In this regard, polyborylated alkenes, which comprise molecules with more than one single C−B bond, would enable greater structural diversity and lead to novel transformations[20,22,24–35]. Therefore, over the last decade, much effort has been expended to synthesize new functionalized classes of polyboronate-alkenes,

[1]Institute of Chemistry, The Center for Nanoscience and Nanotechnology, Casali Center for Applied Chemistry, The Hebrew University of Jerusalem, Jerusalem 9190401, Israel. ✉e-mail: Ahmad.Masarwa1@mail.huji.ac.il

which are excellent building blocks for the modular construction of new compounds[22,33,36–38]. These poly-organometallic-ethene equivalent species can be categorized into four main different types: The *gem*-diborylalkene **I**[39], the vicinal diborylalkene **II**, the triborylalkene **III**, and fully metalated tetraborylethene **IV** (Fig. 1a)[24,39,40]. Recently, these polyborylated-alkenes have shown a large variety of applications, including the synthesis of materials

with aggregation-induced emission properties for potential utilization in biological imaging and chemical sensing[24]. Furthermore, these species e.g., triborylalkenes **III** have been investigated in vitro as potential matrix metalloproteinase inhibitors[22,41]. However, in many cases, their side-selective transformations might somehow be complicated because of the similar behavior of the boryl groups[24,40]. One potential method to overcome this main limitation relies on

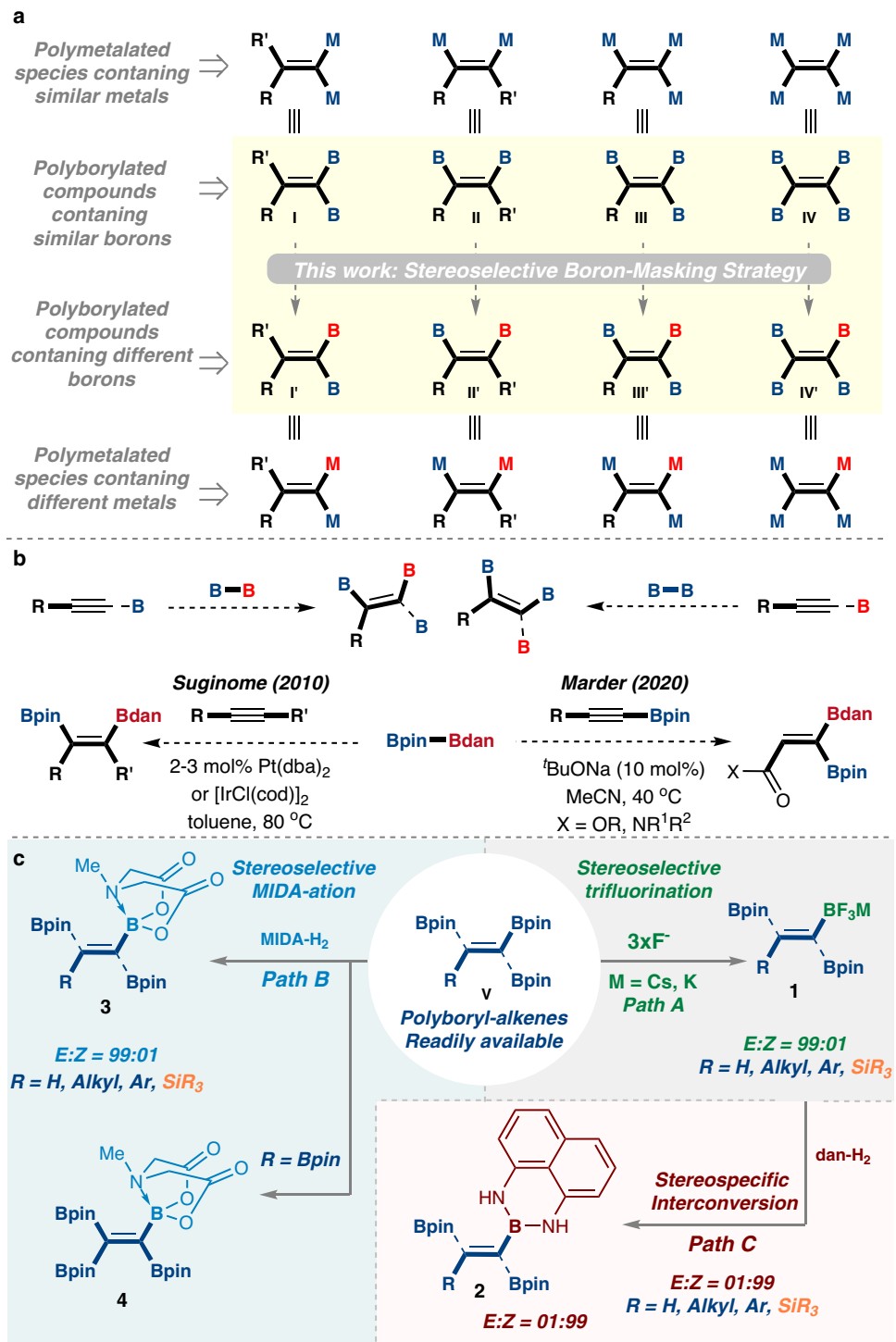

**Fig. 1 | General scheme of the work. a** Classifying of polyborylated and polymetalated compounds. **b** Representative routes for making mixed polyborylated-alkenes. **c** General scheme of the presented approaches for the stereoselective boron-masking of polyborylated-alkenes includes the trifluorination, MIDA-ation, and stereospecific interconversion protocols. B boron group, M metal, Bpin pinacolato-boron, Bdan B-1,8-diaminonaphthalene. BMIDA B-N-methyliminodiacetic acid.

installing a unique boryl group that can then react differently under the appropriate conditions[7,23,42]. In this regard, their ability to be tuned toward different reactivity and transformability can be achieved through the hybridization of borons and by the vacant $p$-orbital of these boryl groups, which are principally based on the choice of the boron-protecting group[22,23,40].

For example, this strategy of protecting-group-dependent reactivity enables the specific synthesis of fully substituted alkenes via selective Suzuki–Miyaura cross-coupling reactions, for instance via protecting groups such as 1,8-diaminonaphthalene (dan) or *N*-methyliminodiacetic acid (MIDA) vs pinacolato (pin)[43,44]. Moreover, we envisioned that having different boryl groups attached directly to the alkene moiety might also tune the reactivity of the double bond toward possible addition-type reactions[7,23,40].

Despite their high synthetic value and wide range of potential applications, there is a limited number of efficient and atom-economical procedures available for the preparation of polyboryl alkenes bearing dissimilar boryl groups[28,29,31,45–47]. Moreover, some of these classes of mixed polyboryl alkenes e.g., **1-4**, cannot be accessible by conventional existing methods (Fig. 1)[45–48].

Traditionally, few of these mixed polyboryl alkenes e.g., **2-3**, can be obtained by using transition-metal catalyzed reactions of (borylated) alkynes using (unsymmetrical) diboranes[49–51] or otherwise (Fig. 1b)[45]. However, because of mechanistic restrictions to *syn*-addition, these methods are limited to provide access to a single isomer. Moreover, some of them e.g., mixed-B-containing tetrasubstituted alkenes, cannot be synthesized by these conventional methods using alkynes (Fig. 1)[48]. In view of this, general and practical methods for accessing such compounds in a stereodefined manner are desirable.

Nevertheless, few methods are currently available. For example, In 2010, Suginome[46] and co-workers reported a practical synthetic method for accessing 1,2 diborylalkenes compounds in a regioselective manner through Pt-catalyzed regioselective 1,2-diboration of alkynes with the unsymmetrical diboron(4)[49–51] reagent Bpin-Bdan in which the Bdan moiety ends up on the terminal carbon (Fig. 1b)[46]. Later, Marder reported a base-catalyzed stereoselective 1,1-diboration of alkynyl esters and amides with Bpin-Bdan to synthesize mixed (*Z*)−1,1-diborylakenes (Fig. 1b)[29]. Inspired by this valuable synthetic methods[29,46] and others[20,22,24,28,32,52] we realized that a complementary and general approach to the synthesis of polyborylalkenes bearing different boryl groups was clearly needed (Fig. 1a–c)[53].

We hypothesized that this could be achieved by a direct transition-metal-free protocol of site-selective and stereoselective boron-masking[23,42,53,54] strategies of the well-established and readily available polyborylated alkenes bearing the same B-groups (**V**) (Fig. 1C).

In this work, the programmed strategies include three designed pathways (Fig. 1c): Path A describes the stereoselective trifluorination masking reaction for the synthesis of the trifluoroborate-alkenes **1** (gray box, Fig. 1c); Path C outlines the stereospecific interconversion of the trifloroborates **1** to the Bdan version **2** (pink box, Fig. 1c); Path B illustrates the stereoselective MIDA-ation masking of the *gem-* and *vic-*diborylalkenes, the triborylated-alkenes, and importantly the tetraborylated alkenes, which furnish the MIDA-ation products **3**, and **4**, respectively. Notably, our general program will lead to efficient synthetic access to families of highly potent polyborylated alkenes possessing mixed boryl groups, with high level of site-selectivity and high stereocontrol (Fig. 1c).

## Results and discussion
### Stereoselective trifluorination masking of (V)
We began our studies by developing a selective, mild, and efficient synthesis for the trifluoroborylated alkene salts **1** via a trifluorination masking protocol of readily available B-containing alkene derivatives [**V-(1-2)**], which undergo nucleophilic-type trifluorination reactions

with MF (Fig. 2a). This approach was inspired by our recent work[55], which mainly explored the desymmetrization of 1,1-diborylalkanes by trifluorination between *gem*-BpinBpin-C(sp$^3$) units and FM salt in the presence of an alkali metal sponge (AMS)[55].

On that basis, we sought to generalize this type of process to include the trifluorination masking reaction of alkenes derivatives [**V-(1-2)**] with FM salts. Key to the success of this general approach was whether trifluoroboryl salts alkenes could be obtained in high site-, chemo-, and diastereoselectivity. A crucial challenge in developing such a masking strategy for these polymetalloids [**V-(1-2)**] lies in preventing the F-nucleophilic addition to the electron-deficient double bond.

Initially, we examined the simple readily available *gem*-diborylethene **V-1** (Fig. 2). We were intrigued to observe that *gem*-di(Bpin) alkene (**V-1a**) could be selectively converted into *gem*-diboryls containing a BF$_3$ moiety through a simple masking fluorolysis process of the Bpin group, at room temperature. The sole product **1a** was observed after a 4–5 min reaction time, using either KF or CsF as salts. The product was obtained by precipitation, thus making product isolation rapid and simple (Fig. 2).

Under the optimized conditions, the scope of the trifluorination reaction was examined (**1a-r**, Fig. 2b). Our results reveal that we can stereoselectively convert *gem*-diborylalkenes[39] possessing alkyl (**1j**), cyclic alkyl (**1k**), vinyl (**1 l**), and ester (**1o**) substituents. In addition, an array of different readily available *gem*-diborylalkenes bearing aromatic rings with electron-withdrawing groups (EWGs) and with electron-donating groups (EDGs), (**1b-1i**), trifluoromethyl groups (**1f**), and heteroatom substituents (**1g**), successfully underwent this trifluorination-type reaction in a highly diastereoselective manner, affording exclusively the *E*-isomer. Generally, the products **1a-1l** were isolated in good yields under the established optimal conditions. Notably, product **1b** was impurely obtained in the presence of KHF$_2$ in a very low yield and after 24 h (See Supplementary Table 1, page 21). Moreover, the selectivity and yield of **1a-b** are not affected when an excess of MF or AMS is used.

The trifluorination masking process also displays selectivity when applied to tetrasubstituted alkenes **1m-n** (Fig. 2b). Importantly, even when both faces of the double bond are only slightly sterically differentiated, for example, Ph vs. Me in alkene **V-1n**, the trifluorination afforded product **1n** with good diastereoselectivity. The structure and relative configuration of the diastereomers of **1n** were unambiguously determined by 2D-NMR NOESY studies (see Supplementary Fig. 294, page 244).

Moreover, when polymetalated alkenes [**V-(2p-r)**] with 1-silyl-1,2-diboryls were used[56], the trifluorination reaction chemoselectively converted the less sterically encumbered boryl site to give trifluoroborate salts **1q-1r** in good yields, keeping the Si-group intact under this fluorination reaction (Fig. 2b). The structure and relative configuration of **1q-r** were simply determined by H-NMR, i.e., the vinylic proton is split into a quartet by the fluorine atoms of the flanking BF$_3$ group (for example see H-NMR-**1r**, Fig. 2b). In contrast, the application of *anti-vic*-1,2-diboryls alkene **V-1t** to the trifluorination reaction conditions resulted in the conversion of both Bpin groups to BF$_3$, yielding product **1t** (Fig. 2b).

Accordingly, we envisioned that the mechanism underlying the selectivity, that is: (1) the observation of only the mono-BF$_3$M product **1**, while the other boron substituent remains untouched, and (2) the observation of the exclusive *E*-isomer, arises from a tandem process in the fluorination step in which the first nucleophilic fluoride attacks the vacant *p*-orbital of the Bpin group on the less sterically hindered face of the double bond of **V-I** (Fig. 2c)[55]. As a result, this fluorinated-boron (see **V-II**) becomes more electrophilic; hence, it forces the second and then the third nucleophilic fluorides to attack only the originally fluorinated-boron center (**V-III**, Fig. 2c)[55]. Consequently, the generated borate (BF$_3$) moiety in (**1**) develops a partial negative charge on the

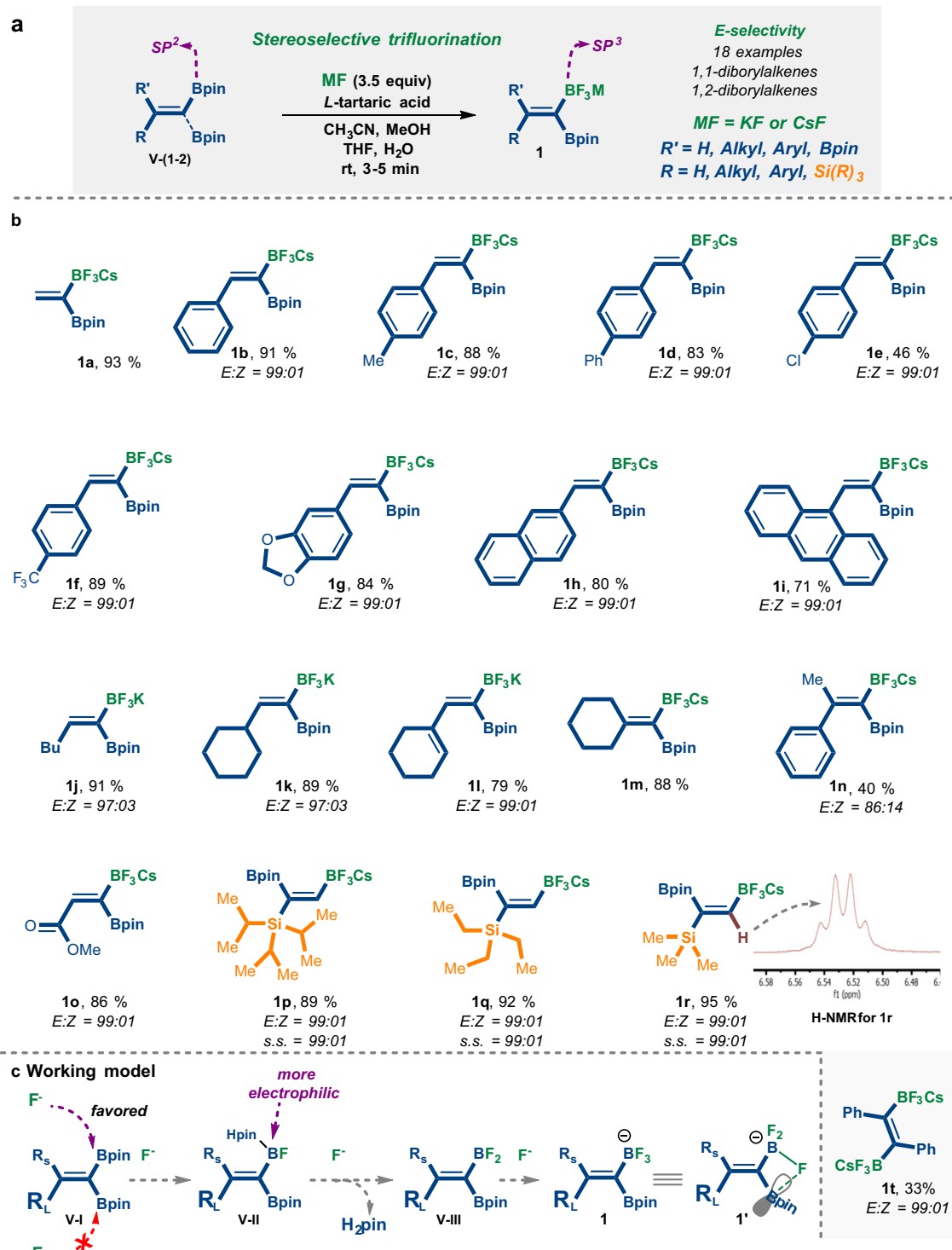

**Fig. 2 | The stereoselective trifluorination Boron-Masking. a** General scheme of the presented trifluorination approach. **b** The scope of the reaction. **c** Working model. Bpin pinacolato-boron. s.s. site-selectivity.

fluorides, and this partial negative charge could stabilize the flanking Bpin group **1** toward subsequent attack by fluoride; this stabilization may occur through a bridging fluoride structure (see **1'**, Fig. 2c)[55].

### Stereospecific interconversion of (1) to product (2)
With these valuable diborylalkenes-BF$_3$M (**1**) in hand, we sought to demonstrate their synthetic utility in the selective transformation of the BF$_3$M group to the Bdan moiety as described in Fig. 3. Unlike the

BF$_3$M moiety, the protected Bdan group is well-known to possess moderated Lewis acidity and to be generally inactive toward trans-metalation, a key step of cross-coupling[7,43,52,53,57].

Notably, only a small number of established methods to prepare this class of *gem*-(Bpin,Bdan)-alkenes e.g., **2** have been reported, and most of them used transition-metal catalysts through a stereoselective 1,1-diboration of alkynes[29,46]. For example, Chirik described a Co-catalyzed 1,1-diboration of aliphatic alkynes to obtain (*Z*)−1,1-

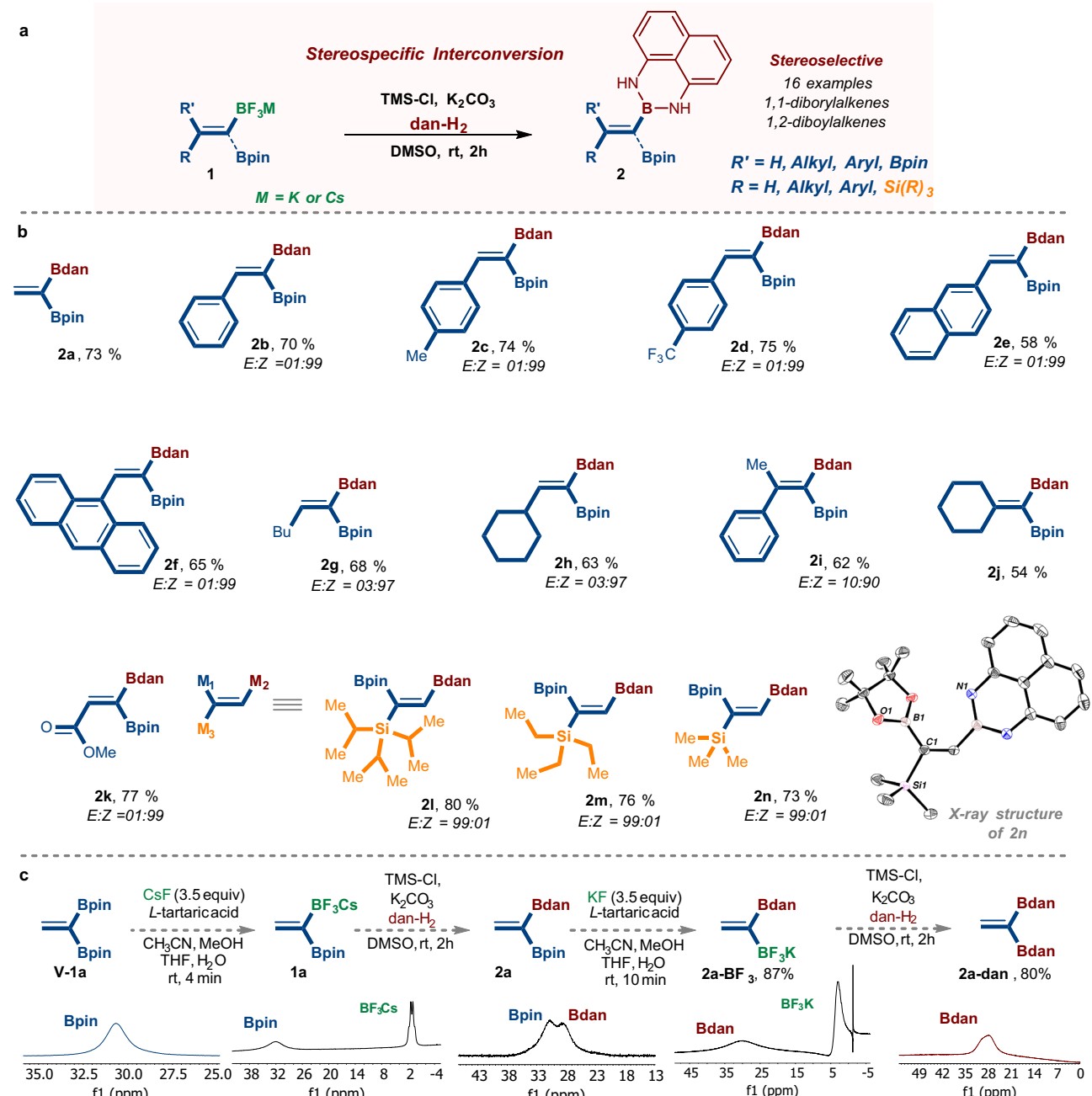

**Fig. 3 | The stereospecific interconversion approach. a** General reaction of the stereospecific interconversion of the triflouroborates into the dan containing alkenes. **b** The scope of the reaction. **c** A sequential B-masking for the preparation of *gem*-di(Bdan)-alkene and [11]B-NMR comparison. Bpin pinacolato-boron, Bdan B-1,8-diaminonaphthalene.

diborylalkenes with the use of mixed diboron reagent Bpin–Bdan[58]. Later, as already stated, Marder, described a base-catalyzed stereo-selective diborylation of alkynyl esters and amides with Bpin–Bdan[29]. In 2020, Engle's group reported a copper-catalyzed method for the *E*-selective 1,1- diborylation of terminal alkynes, through a tandem process that involves sequential dehydrogenative borylation of the alkyne substrate with HBdan, followed by hydroboration with HBpin[57]. Recently, Nguyen reported the use of P1 – ᵗBu phosphazene as a super-basic organocatalyst to promote 1,1-diborylation reactions that are limited to inactivated aromatic and electron-deficient terminal alkynes[59]. These methods, though highly enabling in their own right, have limited substrate scope and provide access to only the trisub-stituted alkenes[59]. Driven by our interest in developing transition-

metal free transformations for accessing the Bpin,Bdan-alkenes bear-ing two substituents at the adjacent carbon position, we envisioned that the masking protocol of the BF₃Cs salts **1** to the Bdan might overcome these limitations (Fig. 3a).

Our investigation commenced by examining reaction condi-tions using the simplest BF₃Cs-containing alkene (**1a**) as a pilot substrate. We were pleased to observe that diborylalkenes (**1a**) could be selectively masked by affording the *gem*-(Bpin,Bdan)-ethene **2a** (Fig. 3b)[55]. Moreover, the *E*-diborylalkenes (**1b-1o**) could be stereospecifically converted, into diboryls bearing the 1,8-dia-minonaphthalene (dan) as the protecting group for the boron moiety (Bdan) through simple solvolysis of organotrifluoroborates **1** in the presence of trimethylsilyl chloride and a bis-nucleophile

1,8-diaminonaphthalene as described in Fig. 3b. Notably, having this mixed (Bpin,Bdan) unit not only will give the flexibility to tune the reactivity of the boron groups, but also might influence the reactivity of the double bond in alkene-Bdan 2[25,60,61].

The same scenario was successfully applied to the trimetalated alkenes 1l-n affording the 1,1-BpinSiR₃-2-Bdans (2l-n) in good yields (Fig. 3b)[46,47,52]. The structure and site selectivity of product 2n was confirmed by X-ray crystallographic analysis (Fig. 3b, See Supplementary Table 5, page 92).

Similarly, we were able to develop a sequential selective masking protocol for the simple *gem*-di(Bpin)ethene V-1a affording the *gem*-di(Bdan)ethene 2a-dan (Fig. 3c). In this protocol, compound V-1a underwent selective fluorolysis generating the *gem*-(Bpin,BF₃Cs) 1a in reasonable yield. Then 1a was selectively converted to the *gem*-(Bpin,Bdan) 2a. Finally, 2a underwent a trifluorination reaction of the Bpin moiety with retention of the Bdan to afford product *gem*-(Bdan,BF₃K) 2a-BF₃. Subsequent treatment with TMS-Cl in the presence of 1,8-diaminonaphthalene gave the corresponding *gem*-di(Bdan)ethene 2a-dan (Fig. 3c). The products of these successive conversions have been clearly determined by the ¹¹B-NMR as described in Fig. 3c.

## Stereoselective MIDA-ation masking of (V)

Encouraged by these results, and to generalize these selective masking strategies for the polyborylated alkenes, we anticipated introducing a valuable type of an *N*-coordinated boronate group, i.e. an *N*-methyliminodiacetic acid (MIDA) protecting group[21,44,62,63], as a masked moiety in the so-called the MIDA-ation reaction (Fig. 4a)[44,53,54].

In this proposed MIDA-ation event, an important feature was the control over the site-, chemo-, and diastereoselectivity when using tri- or tetra-substituted alkenes. If this scenario succeeds to provide the mixed (Bpin,BMIDA)-alkenes, it can indeed represent another rare variety of the polyborylated alkene compounds.

We started our study by applying the MIDA-ation conditions to the symmetrical *gem*-(Bpin,Bpin)- diborylethene V-1a, by a very simple protocol of adding *N*-methyliminodiacetic acid, DMSO as a solvent, and heating the reaction to 110 °C for 24 h. We were pleased to observe that the MIDA-ation masking process could be selectively controlled to forge the mono B-MIDA, the thermodynamically favored product 3a in 63% yield. The structure of product 3a was confirmed by X-ray crystallographic analysis. Interestingly even when an excess of *N*-methyliminodiacetic acid has been used, the reaction still delivers the mono BMIDA product. This might indicate that the mono selectivity is primarily controlled by sterics, in a good agreement with the fact that the BMIDA group is a bulky group, which prevent the installation of an additional bulky MIDA group.

Having these optimized conditions, we started to explore the scope of the protocol, questioning the selectivity when having polyboron-containing alkenes.

When applying the MIDA-masking reaction over the *gem*-diboryl V-1b, the reaction was not only selectively converting the sole Bpin group to the BMIDA unit, but also, the reaction showed an excellent stereoselectivity, affording exclusively the *E*-isomer in a good yield. The *E*-configuration of product 3b was unambiguously determined by X-ray crystallographic analysis and by 2D-NMR NOESY (see Supplementary Table 7, page 94 for X-ray).

By analogy, an array of different readily available *gem*-diborylalkenes V-1 bearing aromatic with EDG and EWG groups (3c-3g), heteroatom (3g), and trifluoromethyl (3f) substituents have been successfully converted to the (Bpin,BMIDA) alkene in an excellent stereoselectivity with good yields. Also, naphthalene (3h), anthracene (3i), and phenanthrene (3j) groups were compatible, giving the same selectivity with good to moderate yields (Fig. 4b). Cyclic alkyls 3m-o and ester group were compatible in this transformation as well,

providing good yields and good to excellent selectivity. Moreover, an additional advantage of this process is that all of these products (3) were purified by simple precipitation only. The MIDA-ation reaction shows selectivity also in the case of *gem*-diBpin-tetrasubstituted alkenes 3k-l (Fig. 4b). Interestingly, even when both faces of the double bond are only slightly sterically differentiated, for example, Ph vs. Me in alkene V-1l, the MIDA-ation afforded product 3l with excellent diastereoselectivity. Based on these results, the masking of the MIDA-ation reaction occurs on the Bpin group at the less hindered side of the double bond, which again supports the observation that the stereoselectivity outcome of this process is most likely controlled by sterics (Fig. 4b).

Next, the polymetalated alkenes (V-2p-r) with 1-silyl-1,2-diboryls were tested (Fig. 4c)[56]. As expected, the MIDA-ation reaction chemoselectively converts the less sterically encumbered boryl site to give (Si,Bpin,BMIDA) alkenes 3p-r. The site selectivity of product 3r was confirmed by X-ray crystallographic analysis (Fig. 4c, see Supplementary Table 8, page 95). Notably, 1,2 bisborylated alkenes were also tested for the MIDA-ation process. Similarly, the reaction showed excellent site-selectivity for the less sterically encumbered boryl site, giving products 3s-3t in good yields (Fig. 4c). Symmetrical tetrasubstituted 1,2-borylated alkenes[30] V-2u still displayed excellent selectivity, affording product 3u with the mono BMIDA, leaving the second boryl unit intact in 75 % yield. Importantly, a slight sterical differentiation on the alkene V-2v still showed an excellent site-selectivity in product 3v with 64% yield (Fig. 4c). Using other 1,2 -C(sp²)-B groups, e.g., 3,4-diboryl diene, yielded product 3w selectively in 46 % yield. When applying the MIDA-ation protocol to the 1,2-bisborylated-ethyne, the reaction showed no selectivity and provided product 3x with two BMIDA groups. These results are in perfect agreement that indeed sterics play a key role in controlling site- and stereoselectivity (Fig. 4c).

Next, we asked ourselves if we could extend these protocols for the tri- and tetra-borylated alkenes V-(3-4) (Fig. 4d, e)[20,22,24,32]. This will lead to a rare class of polyborylated alkenes that contain the BMIDA group. In this regard, in 2013, Nishihara reported a single example of the synthesis of the *Z*-isomer of 1,1,2-triboryl-2-phenylethene having one BMIDA moiety (Ph,Bpin,BMIDA,Bpin)[48]. This was achieved by a platinum-catalyzed diborylation of BMIDA-phenylethynyl with bis-(pinacolato)diboron[48]. To explore a more general and complementary approach to accessing this rare class of materials, we imagined that this might be accomplished by our masking protocol. In the event, when the MIDA-ation conditions were applied to the 1,1,2-triboryl-2-phenylethene V-3y-ab having three Bpin groups[20], the reaction was demonstrated excellent site-, chemo-, and stereoselective levels resulting in access to *E*-isomers 3y-ab, in good yields (Fig. 4d). The configuration of the *E*-isomer was confirmed by X-ray crystallography as shown in 3aa in Fig. 4d.

Most exceptionally, our MIDA-ation scenario was applicable to the tetraborylethene V-4[24], resulting in converting selectively only one of the four identical Bpin groups to the BMIDA-tetraborylated ethene 4 in 69% yield (Fig. 4e). Remarkably, even when the reaction was performed with a huge excess of MIDA (10 equiv) for 2 days, the reaction still yielded product 4 as a sole product. Moreover, when reacting product 4 by applying additional MIDA-ation protocol, no double MIDA-ation was observed. To shed more light on the reasons for the mono-selectivity masking of the tetraborylmethene, X-ray crystallographic analysis was performed for both structures V-4[24] and 4. This gave us the opportunity to compare the angles and relative bonds' distances for better understanding. Accordingly, we observed that after the first masking, i.e., MIDA-ation occurs, the reset of the three Bpin groups move away from the sterically hindered MIDA center and become closer to each other. This results in a crowded structure that might indeed be enough to prevent the MIDA group

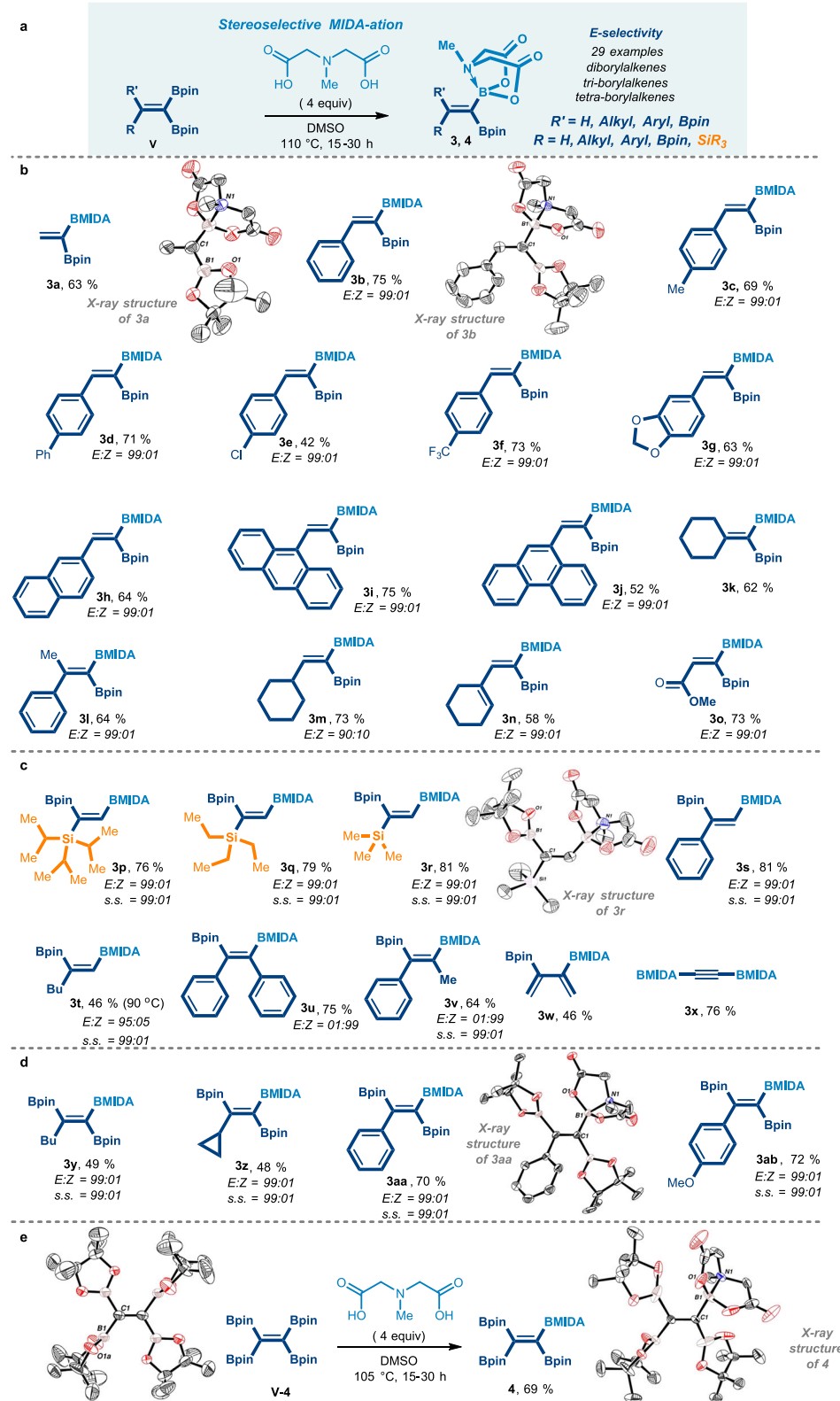

**Fig. 4 | The stereoselective MIDA-ation Boron-Masking strategy. a** General scheme of the presented MIDA-ation reaction. **b** The reaction scope of *gem*-diborylalkenes. **c** The reaction scope of 1,2-diborylalkenes. **d** The reaction scope of 1,1,2-triborylalkenes. **e** B-masking for the tetraborylethene. Bpin pinacolato-boron. Bpin pinacolato-boron, BMIDA B-*N*-methyliminodiacetic acid. S.S. site selectivity.

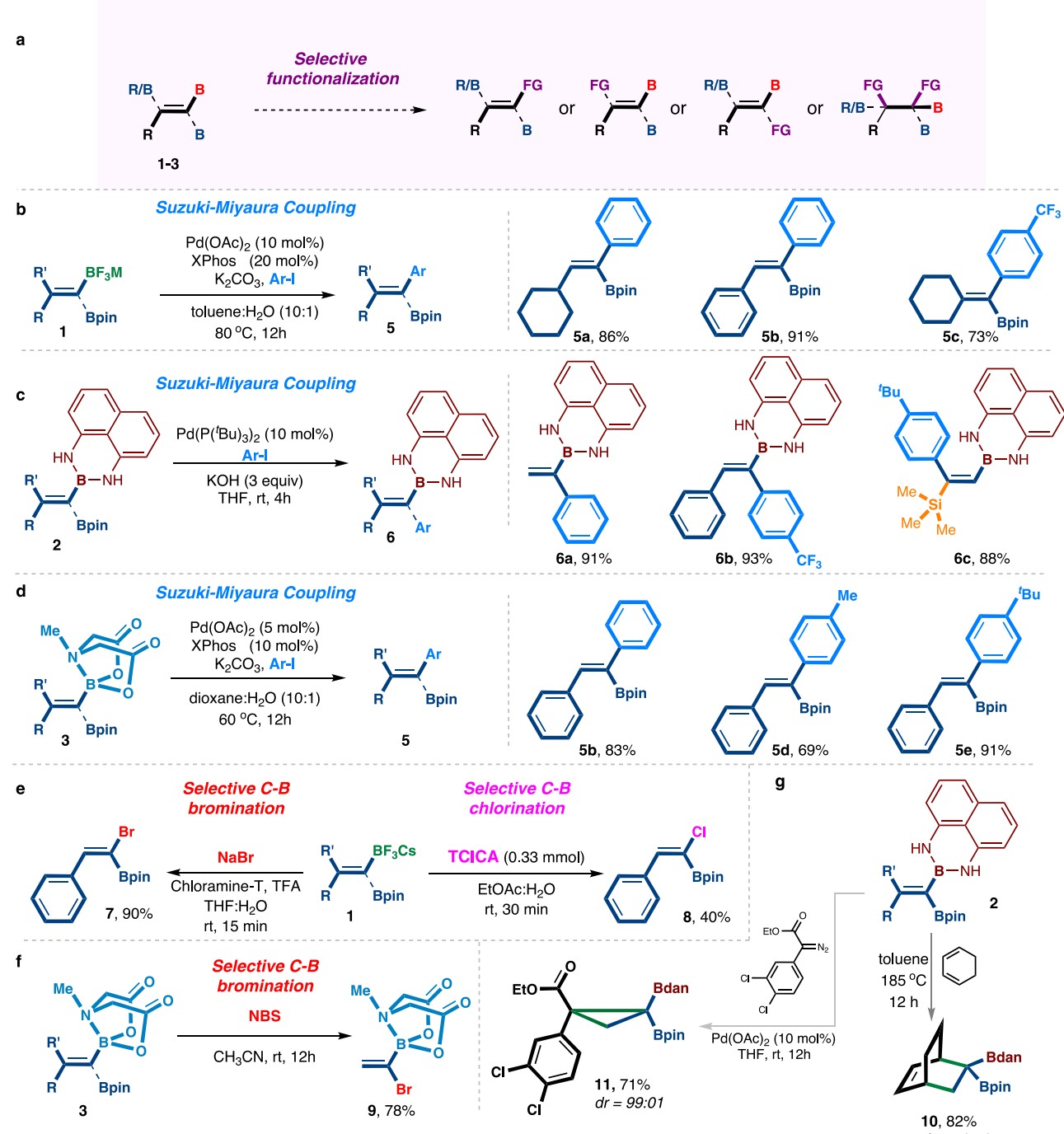

**Fig. 5 | Representative Synthetic utility of polyBorylated-Alkenes (1-3).**
**a** General scheme of the representative applications**. b** Selected examples of
Suzuki−Miyaura Coupling of **1. c** Representative Suzuki-Miyaura Coupling reactions
on **2. d** Selected examples of arylation reactions on **3. e** Presented halogenation

reactions on **1. f** Bromination of the MIDA-ation product **3. g** Cycloaddition reac-
tions of **2**. Bpin pinacolato-boron, Bdan B-1,8-diaminonaphthalene. BMIDA B-N-
methyliminodiacetic acid. TCICA Trichloroisocyanuric acid. NBS *N*-
bromosuccinimide.

from approaching those Bpin groups and avoid the second masking
from occurring (Fig. 4e).

## Synthetic utility of polyborylated-alkenes (1-3)
To showcase the synthetic usefulness of these polymetalloid alkene
products of our masking strategies, we aimed to demonstrate their
synthetic utility in selective transformations of the C−B bonds and
the double bond moiety. To this end, various representative appli-
cations were conceived for some of these valuable polyborylated
alkenes **1**-**3** as depicted in Fig. 5a[28]. For instance, we conducted a

Suzuki−Miyaura cross-coupling reaction of the mixed-borylated
product **1**-**3** with aryl iodides[64]. Accordingly, the diborylalkenes-
BF$_3$M (**1**) reacted selectively to form a new C−C bond from the
C−BF$_3$M bond, providing trans-borylalkenes **5** (Fig. 5b). Trisub-
stituted diborylalkenes-BF$_3$M (**1**) containing alkyl and aryl groups
e.g., **1b**, **1k**, and fully substituted alkene **1m** were also compatible,
affording the desired cross-coupling products **5a**, **5b**, and **5c**
respectively, in good yields[64].

When the mixed Bpin,Bdan-alkenes **2** was submitted to the Suzuki
−Miyaura coupling reaction with aryl-iodide (Fig. 5c), the coupling

occurred at the Bpin part chemoselectively, leaving the Bdan moiety intact in products **6**[57]. This include the simple *gem*-(Bpin,Bdan)-ethene **2a**, the *gem*-(BpinBdan)-alkene **2b**, and particularly the 1,1-BpinSiMe₃-2-Bdans (**2n**). It is worth mentioning that some of these vinyl boronate products e.g., **6a** and **6c** are otherwise not easily accessible[57].

With the synthesized mixed-(Bpin,BMIDA)-compound **3** in hand, we investigated the chemoselective palladium-catalyzed arylation reaction with aryl-iodide. Under these conditions, product **5** was obtained in high chemoselectivity (Fig. 5d). Unlike the diverse reactivity of diborylalkenes-BF₃M **1** in the Suzuki–Miyaura coupling reaction, compounds **3** bearing alkyl groups (e.g., **3m**, **3k**) were found to be unsuitable for the reaction.

Next, we investigated the selective halogenation of **1** and **3** (Fig. 5e, f). In this regard, diborylalkenes-BF₃Cs **1b** underwent chemoselective bromodeboronation[65] and chlorodeboronation[66] under mild reaction conditions with the Bpin moiety remaining untouched (Fig. 5e). Treatment of MIDA-ation product **3a** with *N*-bromosuccinimide (NBS) furnished chemoselectively the monohalo-borylated alkene **9**, leaving the BMIDA group intact (Fig. 5f)[20].

Finally, the unsymmetrical *gem*-diborylalkene **3** was independently subjected to Diels–Alder[25] and the cyclopropanation[60] reactions conditions (Fig. 5g). We obtained the desired mixed *gem*-diborylated cycloaddition products **10** and **11** respectively in good yield as described in Fig. 5g[25,60].

In conclusion, we have designed masking strategies that address the long-standing challenge of site- and stereoselective access to stereodefined polymetalloid alkenes. This was achieved by introducing three protocols that enable the use of the readily available polyBpin-containing alkenes as a reasonably reactive nucleophilic-type acceptor for the masking reactions. This includes: (1) trifluorination masking reaction; (2) the interconversion of the trifluoroborate-alkenes to the Bdan-alkenes; and (3) MIDA-ation reactions. The products of these reactions enable the formal synthesis of polyborylated alkene classes, particularly 1,1-di-, 1,2-di-, 1,1,2-tris-(metalated) alkenes containing BF₃Cs, Bdan, and BMIDA, as well as the 1,1,2-tri- and 1,1,2,2-tetra-borylated alkenes, which would be difficult to access with existing strategies. The setups are simple and can be carried out in a transition-metal-free manner. The products, many of which are valuable building blocks, were obtained in high yields and excellent chemo-, site-, and diastereoselectivity. The mixed polymetalloid alkene products were successfully applied in the selective transformations of the C−B bonds and the olefin moiety with excellent chemo- and stereoselectivity. Studies to achieve further selective transformations of these stereo-defined polyborylated-alkenes are currently under investigation and will be reported in due course.

## Methods

### General Procedure for the preparation of triflourination Products (1)

To initiates the reaction, a diboroalkene substrate [**V-(1-2)**] (0.10 mmol) was added to an air-open flask followed by the addition of a mixture of acetonitrile (0.5 mL) and methanol (0.5 mL). Following that, to this mixture, a solution of fluoride salt i.e. KF or CsF (0.35 mmol) in H₂O (0.2 mL) was added, and stirring was continued for 2 minutes at room temperature. Next, *L*-tartaric acid (0.20 mmol) in THF (0.7 mL) was gradually added to the mixture, which was being stirred vigorously. The stirring was continued for an additional 4 minutes, during which a white precipitate formed. The mixture was filtered to remove the white precipitate and thoroughly washed with excess acetonitrile (5 mL). The filtrate was then concentrated using a reduced-pressure evaporator, resulting in a residue of crude solid. The solid residue was washed with diethyl ether and hexane, giving rise to the corresponding organotrifluoroborate salt alkene (**1**) as an amorphous white solid. The solid was then dried further under high vacuum overnight.

### General procedure for the preparation of *gem*-(Bpin,Bdan)-alkene (2)

An oven-dried 10 ml Schlenk tube was cooled under N₂ before the addition of potassium carbonate (3 equiv, 0.3 mmol) and trifluoroborate salt (**1**) (0.1 mmol). DMSO (1 ml) was then added to the reaction mixture along with 1,8-diaminonaphthalene (1.5 equiv, 0.15 mmol), and following the dropwise addition of trimethylsilyl chloride TMS-Cl (3 equiv, 0.3 mmol), then the mixture was stirred for 2 hours at room temperature. After completion of the reaction, the mixture was diluted with EtOAc (3 ml) and H₂O (3 ml), and the residue was extracted with EtOAc (3 × 5 mL). The organic layers were then washed with brine, dried over MgSO₄, and evaporated by an evaporator under reduced pressure to obtain a crude material. The crude material was further purified by a short column on silica gel, resulting in the formation of a yellow solid, which was identified as product (**2**).

### General procedure for the preparation of *gem*-(Bpin,BMIDA)-alkene (3, 4)

Inside the glove box, to a 10 mL thick-walled reaction tube equipped with a magnetic stirring bar, *gem*-diborylalkene (**V**) (0.1 mmol, 1 equiv) and *N*-methyliminodiacetic acid (MIDA) (0.4 mmol, 4 equiv) were added, followed by 1 ml of DMSO. The tube was sealed with a crimped septum cap, taken out of the glove box and heated at 110 °C for the indicated amount of time (15–30 h). The reaction mixture was then diluted with a dropwise addition of water (1 mL) and extracted with EtOAc (2 × 5 ml). The combined organic phase was washed with brine 10 ml, and dried over magnesium sulfate. The solvents were removed under reduced pressure (by evaporator), forming crude matirial as a solid, that was then washed several times with diethyl ether and hexane, yielding **3**, **4** as a solid.

## Data availability

The data supporting the findings of this study are available within the paper and its Supplementary Information. Details about materials and methods, experimental procedures, characterization data, NMR spectra are available in the Supplementary Information. Crystallographic data for the structures reported in this Article have been deposited at the Cambridge Crystallographic Data Centre, under deposition numbers CCDC 2237708 (**2n**), 2194038 (**3a**), 2194037 (**3b**), 2194039 (**3r**), 2194040 (**3aa**), 2194042 (**V-4**), 2194041 (**4**). Copies of the data can be obtained free of charge via https://www.ccdc.cam.ac.uk/structures/.

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

## Acknowledgements

This research was supported by grants from The U.S.-Israel Binational Science Foundation (Grant No. 2020033) and the Israel Science Foundation (Grant No. 287/21) to A.M. We thank Dr. Benny Bogoslavsky (The Hebrew University of Jerusalem, Israel) for the X-ray structure determination. N.H., are grateful for a fellowship from The Hebrew University of Jerusalem. N.E. is thankful for a Ph.D. fellowship from the Neubauer Foundation. We are grateful to Professor Gary Molander (University of Pennsylvania) for useful discussions. This Article is dedicated to Professor Ilan Marek on the occasion of his 60th birthday.

## Author contributions

N.E. and N.H. planned, conducted and analyzed experiments. A.M. designed and directed the project and wrote the manuscript with contributions by N.E. and N.H. All authors contributed to discussions.

## Competing interests

The authors declare no competing interests.
