## [Peer Review File · Nature Communications]

Stereodefined polyMetalloid Alkenes Synthesis via Stereoselective Boron-Masking of polyBorylated-AlkenesREVIEWER COMMENTS

Reviewer #1 (Remarks to the Author):

The manuscript by Masarwa et al., describes an efficient method to access novel families of polyborylated alkenes possessing mixed boryl groups, with site-selectivity, due to steric issues. The noteworthy results are based on the selective incorporation of BF₃M, BMIDA or Bdan units along 1,1-diborylated or 1,2-diborylated alkenes, as well as tri- or tetra-borylated alkenes. The substrate scope covered is representative and the yields of the polyborylated compounds are also significantly high. That represents an advance towards the differentiation of Bpin units within the same molecule by sterically hindered parameters. Although there are some precedents in the recent literature that explore alternative protocols towards the synthesis of mixed polyborylated alkenes, some of them lack of substrate scope or stereoselectivity.

The methodology is sound and the experimental studies support the claim and conclusions.

The Supporting Information also gives the detailed required data to be reproducible.

This reviewer is missing a potential application towards the functionalization of a representative molecular polyborated compounds 1, 2, 3 and 4, in cross coupling reactions.

In addition compounds type II (cis vicinal diborylalkenes) are stereoselectively transformed, however this reviewer raises the question whether trans vicinal diborylalkenes might also behave with high stereoselectivity.

References are well described but few errors need attention from the authors:

Ref 7 editors are missed

Ref 11, ACS instead of Acs

Together with reference 16 there are two parallel reviews on 2018 in Adv Synth Catal and Tetrahedron based on 1,1-diborylalkanes and their reactivity that also deserve to be included,

Ref 30 ACS instead of Acs

Ref 36 is wrong

Ref 41m ACS instead of Acs

This reviewer supports the publication of this manuscript, considering the previous raised comments and questions.

Elena Fernandez

Reviewer #2 (Remarks to the Author):

This is an interesting and potentially very useful paper. The compounds prepared are all well-characterized including several crystal structures. An interesting and important issue is that silyl groups remain intact under their fluorination conditions, as noted in passing on page 5. However, there are several issues which I believe need to be addressed for it to be suitable for publication. While the routes herein seem quite general, as demonstrated by many examples, and will be useful, there are indeed already some routes to both 1,2- and 1,1-unsymmetric bis(borylated) alkenes. While there are a couple of papers cited regarding this subject by Suginome and Marder, for example, they are generally dismissed as having some limitations. Nonetheless, it might be useful to the reader to see these known routes, and any others which form related unsymmetric polyborylated cpds, displayed in Figure 1. That would put the current work in better perspective. In fact, the routes described by both Suginome and Marder utilize unsymmetrical diboron(4) reagents such as Suginome's pinB-Bdan, in reactions with alkynes. In addition, the authors should cite new routes to unsymmetrical diboron(4) reagents such as those of C. Borner, M. T. Wiecha, C. Kleeberg, *Unsymmetrical Diborane(4) Derivatives: A Comparative Study*, *Eur. J. Inorg. Chem.*, 2017, 4485-4492, and P. M. Rutz, J. Grunenberg, C. Kleeberg, *Organometallics*, 2022, 41, 3044-3054.

In fact, the references need careful checking and cleaning up. First, in the Introduction, I wonder about the choice of the first 3 references. There are numerous reviews on organoboron chemistry. Regarding boronic acids and esters and their applications, the book by Dennis Hall (reference 5, which is not properly cited, as the whole book is relevant, and the publisher and city of publication are missing) would be a good start. There are many papers and reviews on applications of boronic acids

as sugar sensors, a few of which are listed below.

M. Dowlut and D. G. Hall, An Improved Class of Sugar-Binding Boronic Acids, Soluble and Capable of Complexing Glycosides in Neutral Water, *J. Am. Chem. Soc.* 2006, 128, 13, 4226–4227

H. Fang, G. Kaur, and B. Wang, Progress in Boronic Acid-Based Fluorescent Glucose Sensors, *J. Fluorescence*, 2004, 14, 481-489.

K. Lacina, P. Skládal and T. D James, Boronic acids for sensing and other applications - a mini-review of papers published in 2013, *Chem. Central J.* 2014, 8:60.

G. Fang, H. Wang, Z. Bian, J. Sun, A. Liu, H. Fang, B. Liu, Qingqiang Yao, and Z. Wu, Recent development of boronic acid-based fluorescent sensors, *RSC Adv.*, 2018, 8, 29400-29427.

Regarding 'materials', there are also many reviews. It should be noted that 'organoboron compounds' constitutes a wide variety of species, not only boronic acids and esters, and even 3-coordinate ones, such as triarylboranes, have numerous applications, sometimes being synthesized via boronate esters or trifluoroborate salts.

A few such reviews include, but are certainly not limited to the following ones, as there are also many other reviews by M. Wagner, M. Ingleson, S. Yamaguchi, S. Wang, F. Jäkle, etc.:

S.M. Berger and T.B. Marder, "Applications of Triarylborane Materials in Cell Imaging and Sensing of Bio-relevant Molecules such as DNA, RNA, and Proteins," *Mater. Horiz.*, 9, 112-120 (2022). DOI: 10.1039/D1MH00696G.

L. Ji, S. Griesbeck, and T.B. Marder, "Recent Developments in and Perspectives on Three-Coordinate Boron Materials: A Bright Future," *Chem. Sci.*, 8, 846-863 (2017). DOI: 10.1039/C6SC04245G.

C.D. Entwistle and T.B. Marder, "Applications of Three-Coordinate Organoboron Compounds and Polymers in Optoelectronics," *Chem. Mater.*, 16, 4574-4585 (2004).

C.D. Entwistle and T.B. Marder, "Boron Chemistry Lights the Way: Optical Properties of Molecular and Polymeric Systems," *Angew. Chem. Int. Ed. Engl.*, 41, 2927-2931 (2002); *Angew. Chem.*, 114, 3051-3056 (2002).

Then, in the last sentence of the Introduction, the authors provide a reference list as '4, 5, 1, 4, 7, 8, 9, 10, 11, 12, 13' which seems a bit odd, as '1' comes after '5' and '4' is repeated.

In the second paragraph, the sentence 'Recently, these polyborylated-alkenes have shown a large variety of applications, including the synthesis of materials with aggregation-induced emission properties for potential utilization in biological imaging and chemical sensing.' has no references cited. While I appreciate that the journal may limit the number of references, which is less than ideal, papers should cite the original works rather than just reviews by other authors. As such, on page 4, second paragraph, where references 17, 20, and 31 are cited (this list should include reference 18 as well), the key original paper by Suginome should certainly be cited:

N Iwadate and M. Suginome, Differentially Protected Diboron for Regioselective Diboration of Alkynes: Internal-Selective Cross-Coupling of 1-Alkene-1,2-diboronic Acid Derivatives, *J. Am. Chem. Soc.* 2010, 132, 2548–2549.

The following paper is also relevant: A. Verma, R. F. Snead, Y. Dai, C. Slebodnick, Y. Yang, H. Yu, F. Yao, W. L. Santos, Substrate-Assisted, Transition-Metal-Free Diboration of Alkynamides with Mixed Diboron: Regio- and Stereoselective Access to trans-1,2-Vinyldiboronates, *Angew. Chem. Int. Ed.* 2017, 56, 5111–5115.

The proposal that the gem-bis(Bpin) compounds undergo selective fluorination at only one Bpin group because of a possible bridging fluoride moiety (cpd 1' in Figure 2, bottom right) is indeed interesting, and it is unfortunate that the authors do not provide any crystal structures of such compounds, although they have several of the Bpin/B-mida cpds.

On page 7, where Marder's base-catalyzed stereoselective diborylation of alkynyl esters and amides is discussed, the authors should cite reference 18 again, as this is the paper in which that was reported (no reference is currently listed for that statement).

On the top of page 9, I suggest a rewriting of the sentence discussing the 'inversion of configuration of the alkene' as, in fact, the configuration of the alkene itself does not change. Only the formal E/Z-terminology relating to the substituents changes, but not the actual geometry around the double bond. I found this sentence confusing.

Finally, in the reference section itself, there are many errors which need to be corrected. Reference 6 is not cited correctly. The journal is 'Chem' not 'Chem-Us'. Reference 7 is not cited correctly as there are no authors or editors and no city of publication. In reference 27, the word 'Asian' does not take a period at the end, as it is not an abbreviation. Reference 30 is not cited correctly. The journal is 'ACS Omega' (ACS in capital letters and no period after either ACS or Omega). Reference 36 (to the author's own work) is not cited properly. The journal is 'Chem. Eur. J.' and the paper has an 'e' number: e202202748.

In the Supporting Information, the isomers are referred to as diastereomers, but I think that this is not the correct term as there are no chiral centers.

In summary, with the above issues corrected or suitably addressed, I would recommend publication of this interesting paper.

Part-2: Response to Comments by Reviews:

We have addressed all the points that were raised by the reviewers individually as shown below. Each reviewer's comment is shown below *in blue*. All changes (in the text) have been marked through **a yellow background**. All modified figures have been highlighted with **green color**.

We would like to express our sincere gratitude to the reviewers for their valuable suggestions. Their insights were incredibly helpful and allowed us to make significant improvements to our work. We appreciate their time and effort in providing such thoughtful feedback.

Response to Comments by Reviewer #1 on the Manuscript

The manuscript by Masarwa et al., describes an efficient method to access novel families of polyborylated alkenes possessing mixed boryl groups, with site-selectivity, due to steric issues.

The noteworthy results are based in the selective incorporation of BF₃M, BMIDA or Bdan units along 1,1-diborylated or 1,2-diborylated alkenes, as well as tri- or teraborylated alkenes. The substrate scope covered is representative and the yields of the polyborylated compounds are also significantly high. That represents an advance towards the differentiation of Bpin units within the same molecule by sterically hindered parameters. Although there are some precedents in the recent literature that explore alternative protocols towards the synthesis of mixed polyborylated alkenes, some of them lack of substrate scope or stereoselectivity. The methodology is sound and the experimental studies support the claim and conclusions. The Supporting Information also gives the detailed required data to be reproducible.

We gratefully acknowledge Prof. Elena Fernandez's constructive suggestions and valuable comments and have made revisions to address her concerns as listed in detail below. We thank her for the positive evaluation of the work and support.

This reviewer is missing a potential application towards the functionalization of a representative molecular polyborated compounds 1, 2, 3 and 4, in cross coupling reactions.

These are very useful and smart suggested ideas and points, and we agree with the reviewer in this regard. Therefore, during the revision time, we conducted tests on the reactivity of these reagents in both cross-coupling reactions and other applications. Our results successfully demonstrated the utility of these polyborylated alkenes **1-3**, including chemoselective Suzuki–Miyaura cross-coupling (products **5-6**, nine representative examples), chemoselective bromodeboronation (products **7** and **9**), chemoselective chlorodeboronation (product **8**), Diels–Alder cycloaddition (product **10**), and cyclopropanation reactions (product **11**). All of these results have been included in the revised manuscript under the new section “*Synthetic utility of polyBorylated-Alkenes (1-3)*” and of Figure 5 for representative synthetic applications of polyborylated alkenes (See revised manuscript Pages 14-15).

Text and SI revised accordingly. We also attempted investigations on product 4, but they were unsuccessful in terms of selectivity. In addition, the abstract and summary parts were revised accordingly.

In addition, compounds type II (cis vicinal diborylalkenes) are stereoselectively transformed, however this reviewer raises the question whether trans vicinal diborylalkenes might also behave with high stereoselectivity. We thank the reviewer for this suggestion. Therefore, a trans-vicinal diborylalkene **V-1t** was subjected to a trifluorination reaction, yielding the product **1t**, where both Bpin groups were transformed to BF₃Cs.

References are well described but few errors need attention from the authors: Ref 7 editors are missed

We thank the reviewer for pointing out these errors. As suggested, we revised and corrected all references. We fixed the ref. 7 and now it appears as ref. 18 in the revised manuscript.

Ref 11, ACS instead Acs.

We fixed ref. 11 and now it appears as ref. 22.

Together with reference 16 there are two parallel reviews on 2018 in Adv Synth Catal and Tetrahedron based on 1,1-diborylalkanes and their reactivity that also deserve to be included,

Both references were added in the revised manuscript, as ref. 34 and ref. 35.

Ref 30 ACS instead of Acs

We fixed it and now it appears as ref. 44.

Ref 36 is wrong

We fixed it and now it appears as ref. 60.

Ref 41m ACS instead of Acs

We corrected it and now it appears as ref. 59.

This reviewer supports the publication of this manuscript, considering the previous raised comments and questions.

Elena Fernandez

Response to Comments by Reviewer #2 on the Manuscript

This is an interesting and potentially very useful paper. The compounds prepared are all well-characterized including several crystal structures. An interesting and important issue is that silyl groups remain intact under their fluorination conditions, as noted in passing on page 5. However, there are several issues which I believe need to be addressed for it to be suitable for publication.

We appreciate the reviewer's time given to review our manuscript. We thank the reviewer for the positive evaluation of the work and the support. We are grateful to the reviewer for pointing out the following comments, errors, and suggestions, which we have addressed in the revised version of the manuscript and SI.

While the routes herein seem quite general, as demonstrated by many examples, and will be useful, there are indeed already some routes to both 1,2- and 1,1-unsymmetric bis(borylated) alkenes. While there are a couple of papers cited regarding this subject by Suginome and Marder, for example, they are generally dismissed as having some limitations. Nonetheless, it might be useful to the reader to see these known routes, and any others which form related unsymmetric polyborylated cpds, displayed in Figure 1. That would put the current work in better perspective. In fact, the routes described by both Suginome and Marder utilize unsymmetrical diboron(4) reagents such as Suginome's pinB-Bdan, in reactions with alkynes.

We would like to express our gratitude to the reviewer for their helpful suggestions. As recommended, we have included the works of Suginome and Marder in Figure 1b as representative routes for making mixed polyborylated alkenes, and have cited them as references 29 and 46, respectively. These indeed improve our introduction.

In addition, the authors should cite new routes to unsymmetrical diboron(4) reagents such as those of C. Borner, M. T. Wiecha, C. Kleeberg, *Unsymmetrical Diborane(4) Derivatives: A Comparative Study*, *Eur. J. Inorg. Chem.*, 2017, 4485-4492, and P. M. Rutz, J. Grunenberg, C. Kleeberg, *Organometallics*, 2022, 41, 3044-3054.

As suggested, the two papers were cited accordingly in the revised manuscript, and appear as ref. 49 and ref. 50.

In fact, the references need careful checking and cleaning up. First, in the Introduction, I wonder about the choice of the first 3 references. There are numerous reviews on organoboron chemistry. Regarding boronic acids and esters and their applications, the book by Dennis Hall (reference 5, which is not properly cited, as the whole book is relevant, and the publisher and city of publication are missing) would be a good start.

We would like to thank the reviewer for pointing these suggestions out, and we fully agree with the reviewer. We revised and corrected all references, and we have made the necessary changes to our manuscript and have properly cited all relevant sources. Ref. 5 has been updated to appear as ref. 1 in the revised manuscript.

There are many papers and reviews on applications of boronic acids as sugar sensors, a few of which are listed below. M. Dowlut and D. G. Hall, *An Improved Class of Sugar-Binding Boronic Acids, Soluble and Capable of Complexing Glycosides in Neutral Water*, *J. Am. Chem. Soc.* 2006, 128, 13, 4226–4227.

We appreciate the reviewer's help in ensuring the accuracy and completeness of our references. The paper by Hall group was cited accordingly and now appears as ref. 8.

H. Fang, G. Kaur, and B. Wang, *Progress in Boronic Acid-Based Fluorescent Glucose Sensors*, *J. Fluorescence*, 2004, 14, 481-489.

The paper was cited accordingly and now appears as ref. 9.

K. Lacina, P. Skládal T. D James, Boronic acids for sensing and other applications - a mini-review of papers published in 2013, *Chem. Central J.* 2014, 8:60.

We added this paper, which now appears as ref. 10.

G. Fang, H. Wang, Z. Bian, J. Sun, A. Liu, H. Fang, B. Liu, Qingqiang Yao, and Z. Wu, Recent development of boronic acid-based fluorescent sensors, *RSC Adv.*, 2018, 8, 29400-29427.

The paper was cited accordingly, and now appears as ref. 11.

Regarding 'materials', there are also many reviews. It should be noted that 'organoboron compounds' constitutes a wide variety of species, not only boronic acids and esters, and even 3-coordinate ones, such as triarylboranes, have numerous applications, sometimes being synthesized via boronate esters or trifluoroborate salts. A few such reviews include, but are certainly not limited to the following ones, as there are also many other reviews by M. Wagner, M. Ingleson, S. Yamaguchi, S. Wang, F. Jäkle, etc.:

We thank the reviewer for these adjustments. All of them have been addressed as the following:

S.M. Berger and T.B. Marder, "Applications of Triarylborane Materials in Cell Imaging and Sensing of Bio-relevant Molecules such as DNA, RNA, and Proteins," *Mater. Horiz.*, 9, 112-120 (2022). DOI: 10.1039/D1MH00696G.

The paper was cited accordingly and now appears as ref. 12.

L. Ji, S. Griesbeck, and T.B. Marder, "Recent Developments in and Perspectives on Three-Coordinate Boron Materials: A Bright Future," *Chem. Sci.*, 8, 846-863 (2017). DOI: 10.1039/C6SC04245G.

This paper was cited accordingly, which now appears as ref. 13.

C.D. Entwistle and T.B. Marder, "Applications of Three-Coordinate Organoboron Compounds and Polymers in Optoelectronics," *Chem. Mater.*, 16, 4574-4585 (2004).

The paper was cited accordingly and now appears as ref. 14.

C.D. Entwistle and T.B. Marder, "Boron Chemistry Lights the Way: Optical Properties of Molecular and Polymeric Systems," *Angew. Chem. Int. Ed. Engl.*, 41, 2927-2931 (2002); *Angew. Chem.*, 114, 3051-3056 (2002).

The paper was cited accordingly and now appears as ref. 15.

Then, in the last sentence of the Introduction, the authors provide a reference list as '4, 5, 1, 4, 7, 8, 9, 10, 11, 12, 13' which seems a bit odd, as '1' comes after '5' and '4' is repeated.

Corrected accordingly.

In the second paragraph, the sentence 'Recently, these polyborylated-alkenes have shown a large variety of applications, including the synthesis of materials with aggregation-induced emission properties for potential utilization in biological imaging and chemical sensing.' has no references cited.

We thank the reviewer for pointing out this issue. It was fixed and now ref. 24 was added accordingly.

While I appreciate that the journal may limit the number of references, which is less than ideal, papers should cite the original works rather than just reviews by other authors. As such, on page 4, second paragraph, where references 17, 20, and 31 are cited (this list should include reference 18 as well), the key original paper by Suginome should certainly be cited:

N Iwadate and M. Suginome, Differentially Protected Diboron for Regioselective Diboration of Alkynes: Internal-Selective Cross-Coupling of 1-Alkene-1,2-diboronic Acid Derivatives, *J. Am. Chem. Soc.* 2010, 132, 2548–2549.

We agree with the reviewer, ref. 18 (that is now ref. 29) was added to the revised manuscript, also the key original paper was also added and now appears as ref. 46.

The following paper is also relevant: A. Verma, R. F. Snead, Y. Dai, C. Slebodnick, Y. Yang, H. Yu, F. Yao, W. L. Santos, Substrate-Assisted, Transition-Metal-Free Diboration of Alkynamides with Mixed Diboron: Regio- and Stereoselective Access to trans-1,2-Vinyldiboronates, *Angew. Chem. Int. Ed.* 2017, 56, 5111–5115.

Santos's paper was added as suggested and now appears as ref. 47.

The proposal that the gem-bis(Bpin) compounds undergo selective fluorination at only one Bpin group because of a possible bridging fluoride moiety (cpd 1' in Figure 2, bottom right) is indeed interesting, and it is unfortunate that the authors do not provide any crystal structures of such compounds, although they have several of the Bpin/B-mida cpds

During the revisions time, we attempted to obtain suitable crystals for these trifluoroborate salts, unfortunately, we could not get the crystals. Alternatively, we were able to obtain the X-ray structure of the BpinSiR₃-2-Bdans compound (**2n**) and we added the structure to Figure 3b and Supplementary Table 5, page S92.

On page 7, where Marder's base-catalyzed stereoselective diborylation of alkynyl esters and amides is discussed, the authors should cite reference 18 again, as this the paper in which that was reported (no reference is currently listed for that statement).

The paper has now been cited in the revised manuscript and appears as ref. 29 on page 8.

On the top of page 9, I suggest a rewriting of the sentence discussing the 'inversion of configuration of the alkene' as, in fact, the configuration of the alkene itself does not change. Only the formal E/Z-terminology relating to the substituents changes, but not the actual geometry around the double bond. I found this sentence confusing.

We are thankful to the reviewer for noticing this issue and agree with the reviewer. We have revised this sentence and now it appears as "Notably, having this mixed Bpin,Bdan unit not only will give the flexibility to tune the reactivity of the boron groups but also might influence the reactivity of the double bond in alkene-Bdan **2**."

Finally, in the reference section itself, there are many errors which need to be corrected Reference 6 is not cited correctly. The journal is 'Chem' not 'Chem-U's'.

Corrected and now appears as ref. 17.

Reference 7 is not cited correctly as there are no authors or editors and no city of publication.

Corrected and now appears as ref. 18.

In reference 27, the word 'Asian' does not take a period at the end, as it is not an abbreviation.

Corrected correctly and now appears as ref. 40.

Reference 30 is not cited correctly. The journal is 'ACS Omega' (ACS in capital letters and no period after either ACS or Omega).

Fixed accordingly and now appears as ref. 44.

Reference 36 (to the author's own work) is not cited properly. The journal is 'Chem. Eur. J.' and the paper has an 'e' number: e202202748.

As suggested, the reference was cited correctly and now appears as ref. 60.

In the Supporting Information, the isomers are referred to as diastereomers, but I think that this is not the correct term as there are no chiral centers.

We thank the reviewer for this valuable comment. In our revised SI, we fixed it and we have now referred to all isomers using the E/Z nomenclature system, and the diastereomers term has been removed.

In summary, with the above issues corrected or suitably addressed, I would recommend publication of this interesting paper.

REVIEWERS' COMMENTS

Reviewer #1 (Remarks to the Author):

The revised manuscript by Masarwa et al., has considered all the suggestions by this reviewer and the new work complements the quality of the paper. This reviewer accept the revised version and confirms the excellence of the work presented.

Reviewer #2 (Remarks to the Author):

The authors have carefully revised their manuscript in accord with the comments of both referees. I am happy to recommend publication of this interesting and potentially very useful paper. There are still a few typos and related issues which should be corrected as listed below.

1) In Figure 3c, for the reaction leading from 1a to 2a, and also 2a-BF₃ to 2a-dan, the letter K in K₂CO₃ should be capitalized, and in the next to last line of the caption for that Figure, there is an 'l' in '-lalkene' which I think should be removed.

2) In Figure 4c, for cpd 3x, I suggest fixing 'ADIMB' at the left side of the structure of the alkyne to 'BMIDA' to be consistent. This is simply a ChemDraw issue. Also, in line 3 of the caption for that Figure, the 'T' in '1,1,2-Triborylalkenes' should not be capitalized.

3) On page 13, second paragraph, the 'B' in 'bis-(pinacolato)diBoron' should not be capitalized.

4) The format for reference 18 (a book) should be corrected to be consistent with that in reference 1, and the city of publication should be added.

5) In reference 40, as noted previously, there should not be a . after 'Asian' as it is not an abbreviation.

Response to Comments by Reviews:

We have addressed a few typos that were raised by the Reviewer-2 as shown below. Each reviewer's comment is shown below *in blue*. All changes (in the text) have been marked through *a yellow background*. All modified figures have been highlighted with *green color*.

Response to Comments by Reviewer #1 on the Manuscript

The revised manuscript by Masarwa et al., has considered all the suggestions by this reviewer and the new work complements the quality of the paper. This reviewer accept the revised version and confirms the excellence of the work presented.

We gratefully acknowledge and thank the reviewer for the positive evaluation of our work and the support for its publication.

Response to Comments by Reviewer #2 on the Manuscript

The authors have carefully revised their manuscript in accord with the comments of both referees. I am happy to recommend publication of this interesting and potentially very useful paper. There are still a few typos and related issues which should be corrected as listed below.

We appreciate the time that the reviewer has dedicated to reviewing our manuscript. We would like to express our gratitude for their positive evaluation of our work and their support for its publication. Additionally, we are thankful to the reviewer for bringing the following typos and errors to our attention, which we have addressed in the revised version of the manuscript.

1) In Figure 3c, for the reaction leading from 1a to 2a, and also 2a-BF₃ to 2a-dan, the letter K in K₂CO₃ should be capitalized, and in the next to last line of the caption for that Figure, there is an 'T' in '-Ialkene' which I think should be removed.

Corrected.

2) In Figure 4c, for cpd 3x, I suggest fixing 'ADIMB' at the left side of the structure of the alkyne to 'BMIDA' to be consistent. This is simply a ChemDraw issue. Also, in line 3 of the caption for that Figure, the 'T' in '1,1,2-Triborylalkenes' should not be capitalized.

Fixed accordingly.

3) On page 13, second paragraph, the 'B' in 'bis-(pinacolato)diBoron' should not be capitalized.

Fixed.

4) The format for reference 18 (a book) should be corrected to be consistent with that in reference 1, and the city of publication should be added.

Corrected.

5) In reference 40, as noted previously, there should not be a . after 'Asian' as it is not an abbreviation.

Corrected.